# Learning Audio-Visual Dereverberation

## Abstract

Reverberation from audio reflecting off surfaces and objects in the environment not only degrades the quality of speech for human perception, but also severely impacts the accuracy of automatic speech recognition. Prior work attempts to remove reverberation based on the audio modality only. Our idea is to learn to dereverberate speech from audio-visual observations. The visual environment surrounding a human speaker reveals important cues about the room geometry, materials, and speaker location, all of which influence the precise reverberation effects in the audio stream. We introduce Visually-Informed Dereverberation of Audio (VIDA), an end-to-end approach that learns to remove reverberation based on both the observed sounds and visual scene. In support of this new task, we develop a large-scale dataset that uses realistic acoustic renderings of speech in real-world 3D scans of homes offering a variety of room acoustics. Demonstrating our approach on both simulated and real imagery for speech enhancement, speech recognition, and speaker identification, we show it achieves state-of-the-art performance and substantially improves over traditional audio-only methods.

## 1 Introduction

Audio reverberation occurs when multiple reflections from surfaces and objects in the environment build up then decay, altering the original audio signal. While reverberation bestows a realistic sense of spatial context, it also can degrade a listener's experience. In particular, the quality of human speech is greatly affected by reverberant environments—as illustrated by how difficult it can be to parse the words of a family member speaking loudly from another room in the house, a tour guide describing the artwork down the hall of a magnificent cavernous cathedral, or a colleague participating in a Zoom call from a cafe. Similarly, automatic speech recognition (ASR) suffers when given reverberant speech input (Kinoshita et al., 2016; Szöke et al., 2019; Watanabe et al., 2020). Thus there is great need for *dereverberation* algorithms that can strip away reverb effects for speech enhancement, recognition, and other downstream tasks, which could in turn benefit many applications in teleconferencing, assistive hearing devices, augmented reality, and video indexing.

The audio community has made steady progress on speech dereverberation (Han et al., 2015; Wu et al., 2016; Zhao et al., 2019b). While past approaches have tackled the problem with signal processing and statistical techniques (Nakatani et al., 2010; Naylor & Gaubitch, 2010), many modern approaches use neural networks to learn a mapping from reverberant to clean spectrograms (Han et al., 2015; Ernst et al., 2018; Fu et al., 2019). To our knowledge, all existing models for dereverberation rely purely on audio. This underconstrains the dereverberation task since the latent parameters of the recording space are not discernible from the audio alone.

However, we observe that in many practical settings of interest—video conferencing, augmented reality, Web video indexing—reverberant audio is naturally accompanied by a visual (video) stream. Importantly, the visual stream offers valuable cues about the room acoustics affecting reverberation: where are the walls, how are they shaped, where is the human speaker, what is the layout of major furniture, what are the room's dominant materials (which affect absorption, reflection, and refraction), and even what is the facial appearance and/or body shape of the person speaking (since body shape determines the acoustic properties of a person's speech, and reverb time is frequency dependent). For example, reverb is typically stronger when the speaker is further away; speech is more reverberant in a large church or hallway; heavy carpet absorbs more sound. See Figure 1.

Our idea is to learn to dereverberate speech from audio-visual observations. In this task, the input is reverberant speech and visual observations of the environment surrounding the human speaker, and the output is a prediction of the clean source audio. To tackle this problem, there are two key

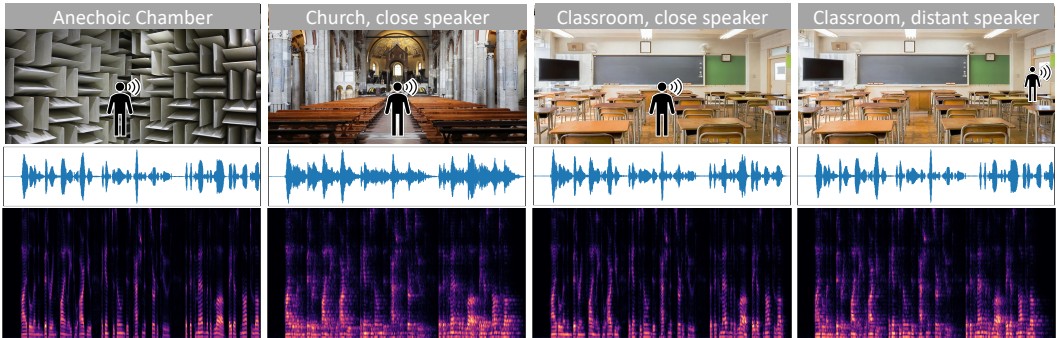

Figure 1: Visual cues reveal key factors influencing reverb effects on human speech audio. For example, these audio speech samples (depicted as waveforms and spectrograms) are identical lexically, but have very different reverberation properties owing to their differing environments. In the church, reverb is strong, in the classroom it is less, and when the speaker is distant from the camera it is again more evident. We propose *audio-visual dereverberation* to learn from both modalities how to strip away reverb effects, thereby enhancing speech quality, recognition, and speaker identification.

technical challenges. First, how to model the multi-modal dereverberation process in order to infer the latent clean audio. Second, how to secure appropriate training data spanning a variety of physical environments for which we can sample speech with known ground truth (non-reverberant, anechoic) audio. The latter is non-trivial because ordinary audio/video recordings are themselves corrupted by reverberation but lack the ground truth source signal we wish to recover.

For the modeling challenge, we introduce an end-to-end approach called Visually-Informed Dereverberation of Audio (VIDA). VIDA consists of a Visual Acoustics Network (VAN) that learns reverberation properties of the room geometry, object locations, and speaker position. Coupled with a multi-modal UNet dereverberation module, it learns to remove the reverberations from a single-channel audio stream. In addition, we propose an audio-visual (AV) matching loss to enforce consistency between the visually-inferred reverberation features and those inferred from the audio signal. We leverage the outputs of our model for multiple downstream tasks: speech enhancement, speech recognition, and speaker verification.

Next, to address the training data challenge, we develop a new large-scale dataset using SoundSpaces (Chen et al., 2020), a 3D simulator for real-world scanned environments that allows both visual and acoustic rendering. We insert "clean" audio voices together with a 3D humanoid model at various positions within an array of indoor environments, then sample the images and properly reverberating audio when placing the receiver microphone and camera at other positions in the same house. This strategy allows sampling realistic audio-visual instances coupled with ground truth raw audio to train our model, and it has the added benefit of allowing controlled studies that vary the parameters of the capture setting. As we will show, the data also supports sim2real transfer.

We first train and evaluate our model on 82 large-scale environments—each a multi-room home containing a variety of objects—coupled with speech samples from the LibriSpeech dataset (Panayotov et al., 2015), then further test with real-world data. We consider both near-field and far-field settings where the human speaker is in-view or quite far from the camera, respectively. The proposed model outperforms methods restricted to the audio stream, improves the state of the art for multiple tasks with speech enhancement, and transfers successfully to real-world data.

Our main contributions are to 1) present the task of audio-visual dereverberation, 2) address it with a new multi-modal modeling approach and a novel reverb-visual matching loss, 3) provide a benchmark evaluation framework built on both SoundSpaces and real data, and 4) demonstrate the utility of AV dereverberation for multiple practical tasks. Overall, our work shows the potential for speech enhancement models to leverage room acoustics by seeing the 3D environment.

## 2 RELATED WORK

**Audio dereverberation and speech enhancement.**  Audio dereverberation and speech enhancement have a long and rich literature (Neely & Allen, 1979; Miyoshi & Kaneda, 1988; Naylor & Gaubitch, 2010; Kinoshita et al., 2016). While dereverberation can be done with microphone arrays, we focus on single audio channel approaches, which require fewer assumptions about the input

data. Recent deep learning methods achieve promising results to dereverberate (Han et al., 2015; Wu et al., 2016; Zhao et al., 2019b; Su et al., 2020b), denoise (Xu et al., 2019; Fu et al., 2019; Su et al., 2020b), or separate (Hershey et al., 2016; Stoller et al., 2018) the audio stream using audio input alone, which can improve downstream speech recognition (Ko et al., 2017; Kinoshita et al., 2016; Watanabe et al., 2020) and speaker recognition (Snyder et al., 2018). Acoustic simulations can provide data augmentation during training (Kinoshita et al., 2016; Ko et al., 2017; Zhao et al., 2020). Some work targets "room-aware" deep audio features capturing reverberation properties (e.g., RT60) (Giri et al., 2015), or synthesizes reverberation effects via acoustic matching (Su et al., 2020a). To our knowledge, the only prior work drawing on the *visual* stream to infer dereverberated audio is limited to using lip regions on near-field faces to first separate out distractor sounds (Tan et al., 2020), and does not model anything about the visual scene for dereverberation purposes. In contrast, our model accounts for the full visual scene, far-field speech sources, and even out-of-view speakers. Our approach is the first to learn visual room acoustics for dereverberation, and it yields state-of-the-art results with direct benefits for multiple downstream tasks.

**Visual understanding of room acoustics.** The room impulse response (RIR) is the transfer function capturing the room acoustics for arbitrary source stimuli; once convolved with a sound waveform, it produces the sound of that source in the context of the particular physical space. While traditionally measured with specialized equipment in the room itself (Stan et al., 2002; Holters et al., 2009) or else simulated with sound propagation models (Allen & Berkley, 1979; Chen et al., 2020; Murphy et al., 2007), recent work explores estimating an RIR from an input image using CNNs (Kon & Koike, 2019) or conditional GANs (Singh et al., 2021) in order to simulate reverberant sound for a given environment. Video-based methods can lift monaural audio into its spatialized (binaural, ambisonic) counterpart in order to create an immersive audio experience for human listeners (Morgado et al., 2018; Gao & Grauman, 2019; Li et al., 2018).

Such methods share our interest in learning the visual properties of a scene that influence the audio channel. However, unlike any of the above methods, rather than generate spatialized audio to benefit human listeners in augmented or virtual reality, our goal is to dereverberate audio— removing the effects of the room acoustics—to benefit automatic speech analysis. In addition, prior methods use imagery taken at camera positions at an unknown offset from the microphone, i.e., conflating all RIRs for a scene with one image, which limits them to a coarse characterization of the environment (Marco Jeub, 2009; Jeub et al., 2010; Murphy & Shelley, 2010). In contrast, our data and model align the camera and microphone to capture novel fine-grained audio-visual properties, including the human speaker's location with respect to the microphone when the speaker is in view.

**Audio-visual embodied learning in realistic simulations.** Recent work in embodied AI explores how vision and sound together can help agents move intelligently in 3D environments. Driven in part by new tools for audio-visual (AV) simulations in realistic scanned environments (Chen et al., 2020), new research develops deep reinforcement learning approaches to train agents to navigate to sounding objects (Chen et al., 2020; Gan et al., 2020), explore unmapped environments (Dean et al., 2020), or move around to better separate multiple overlapping sounds in a house (Majumder et al., 2021). Our work also leverages state-of-the-art AV simulations for learning, but our objective and models are entirely different. Rather than train virtual robots to move intelligently, our aim is to clean reverberant audio for better speech analysis.

**Audio-visual learning from video.** Multi-modal video understanding has experienced a resurgence of work in the vision, audio, and machine learning literature in recent years. This includes exciting advances in self-supervised cross-modal feature learning from video (Morgado et al., 2020; Alwassel et al., 2020; Korbar et al., 2018), localizing objects in video with both sight and sound (Hu et al., 2020), and audio-visual speech enhancement or separation (Ephrat et al., 2018; Owens & Efros, 2018; Zhao et al., 2019a; Afouras et al., 2018; 2020; Michelsanti et al., 2020; Sadeghi et al., 2020; Zhou et al., 2019; Hou et al., 2017). None of these methods address speech deverberation.

## 3 THE AUDIO-VISUAL DEREVERBERATION TASK

We introduce the novel task of *audio-visual dereverberation*. In this task, a speaker (or other sound source) and a listener are situated in a 3D environment, such as the interior of a house. The speaker— whose location is unknown to the listener—produces a speech waveform $A_s$. A superposition of the direct sound and the reverb is captured by the listener, denoted $A_r$. The reverberant speech $A_r$ can be modeled as the convolution of the anechoic source waveform $A_s$ with the room impulse response

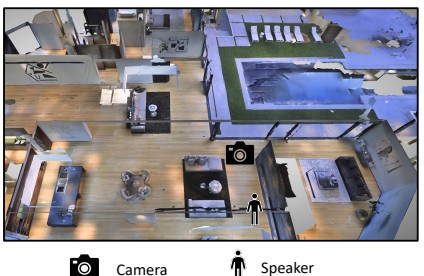 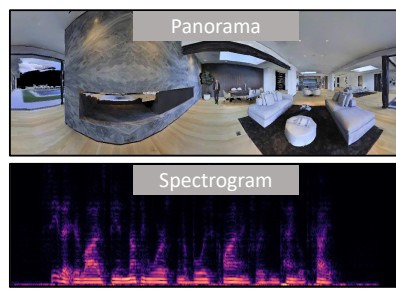

Figure 2: Audio-visual rendering for a Matterport environment. Left: bird's-eye view of the 3D environment. Right: panorama image rendered at the camera location and the corresponding received spectrogram.

(RIR) $R$, i.e. $A_r(t) = A_s(t) * R(t)$ (Neely & Allen, 1979). $R$ is a function of the environment's geometry, its materials, and the positioning of the speaker and the listener. It is possible in principle to measure the RIR $R$ for a real-world environment, but doing so can be impractical when the source and listener are able to move around or we must cope with different environments. Furthermore, in the common scenario where we want to process video captured in environments to which we have no physical access, measuring the RIR is simply impossible.

Crucial to our task, we consider an alternative source of information about the environment: vision. We assume the listener observes its surroundings from an RGB-D camera or an RGB camera coupled with single-image depth estimation (Eigen et al., 2014; Godard et al., 2019). Intuitively, we should be able to leverage the information about the environment's geometry and material composition that is implicit in the visual stream—as well as the location of the speaker (if visible)—to estimate its reverberant characteristics. We anticipate that these cues can inform an estimate of the room acoustics, and thus the clean source waveform. Given the RGB $I_r$ and depth image $I_d$ captured by the listener from its current vantage point, the task is to predict the source waveform $A_s$ from the images and reverberant audio: $\hat{A}_s(t) = f_p([I_r, I_d, A_r(t)])$. This setting represents common real-world scenarios previously discussed, and poses new challenges for speech enhancement and recognition.

## 4 DATASET CURATION

For the proposed task, obtaining the right training data is itself a challenge. Existing video data contains reverberant audio but lacks the ground truth anechoic audio signal, and existing RIR datasets (Marco Jeub, 2009; Jeub et al., 2010; Murphy & Shelley, 2010) do not have images paired with the microphone position. We introduce both real and simulated datasets to enable reproducible research on audio-visual deverberation.

**3D environments and audio simulator.** First we introduce a large-scale dataset in which we couple real-world visual environments with state-of-the-art audio simulations accurately capturing the environments' spatial effects on real samples of recorded speech. We want our dataset to allow control of a variety of physical environments, the positions of the listener/camera and sources, and the speech content of the sources—all while maintaining both the observed reverberant $A_r(t)$ and ground truth anechoic $A_s(t)$ sounds. To this end, we leverage the audio-visual simulator SoundSpaces (Chen et al., 2020), which provides precomputed RIRs $R(t)$ on a uniform grid of resolution 1 m for the real-world environment scans in Replica (Straub et al., 2019) and Matterport3D (Chang et al., 2017). Material modeling is realized by mapping object classes to their reflection and absorption coefficients. We use 82 Matterport environments due to their greater scale and complexity; each environment has multiple rooms spanning on average 517 m$^2$. The rooms' visual and geometric variety offers an opportunity to learn natural acoustic associations (see distribution of reverberation times in Supp.).

**Reverberant speech in 3D visual environments.** We extend SoundSpaces to construct reverberant speech. As the source speech corpus we use LibriSpeech (Panayotov et al., 2015), which contains 1,000 hours of 16kHz English speech from audio books. We train our models with the train-clean-360 split, and use the dev-clean and test-clean sets for validation and test splits, respectively. Note that these splits have non-overlapping speaker identities. Similarly, we use the standard disjoint train/val/test splits for the Matterport 3D visual environments (Chen et al., 2020). Thus, neither the houses nor speaker voices observed at test time are ever observed during training.

For each source utterance, we randomly sample a source-receiver location pair in a random environment, then convolve the speech waveform $A_s(t)$ with the associated SoundSpaces RIR $R(t)$ to obtain

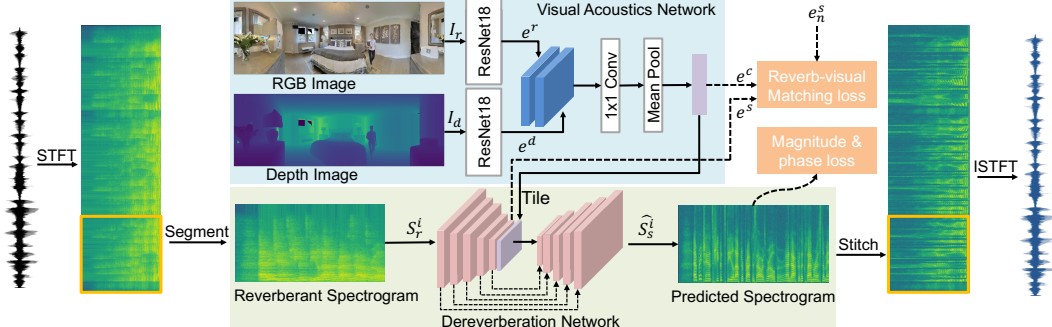

Figure 3: VIDA model architecture. We convert the input speech to a spectrogram and use overlapping sliding windows to obtain 2.56 second segments. For visual inputs, we use separate ResNet18 networks to extract features $e^r$ and $e^d$, which are fused to obtain $e^c$. We feed the spectrogram segment $S_r^i$ to a UNet encoder, tile and concatenate $e^c$ with the encoder's output, then use the UNet decoder to predict the clean spectrogram $\hat{S}_s^i$. During inference, we stitch the predicted spectrograms into a full spectrogram and use Griffin-Lim (Griffin & Lim, 1984) to reconstruct the dereverberated waveform.

the reverberant $A_r(t)$. To augment the visual stream, we insert a 3D humanoid of the same gender as the real speaker at the speaker location and render RGB and depth images at the listener location. We consider two types of image: panorama and normal field of view (FoV). For the panorama image, we stitch 18 images each having a horizontal FoV of 20 degrees, for a full image resolution of $192 \times 756$. For the normal FoV, we render images with a 80 degree FoV, at a resolution of $384 \times 256$. While the panorama gives a fuller view of the environment and thus should allow the model to better estimate the room acoustics, the normal FoV is more common in existing video and thus will facilitate our model's transfer to real data. See Fig. 2. We generate 49,430/2,700/2,600 such samples for the train/val/test splits, respectively. See Supp. materials for examples and details.

**Real data collection.** To explore whether models trained in simulation can also work in the real world, we also collect a set of real images and speech recordings while preserving the ground truth anechoic audio. To collect image data, we use an iPhone 11 camera to capture a panoramic RGB image and a monocular depth estimation algorithm (Godard et al., 2019) to generate the corresponding depth image. To record the audio, we use a ZYLIA ZM-1 microphone. For the source speech, we play utterances from the LibriSpeech test set through a loudspeaker held by a person facing the camera. We collect data from varying environments—auditoriums, meeting rooms, atriums, corridors, and classrooms. For each environment, we vary the speaker location from near-field to mid-field to far-field. For each location, we play 10 utterances of different genders and speakers. During data collection, the microphone also records ambient sounds like people chatting, door opening, AC humming, etc. In total, we collect 200 recordings. Please see the Supp. for examples. We will publicly share all data and code.

## 5   APPROACH

We propose the Visually-Informed Dereverberation of Audio (VIDA) model, which leverages visual cues to learn representations of the environmental acoustics and sound source locations to dereverberate audio. While our model is agnostic to the audio source type, we focus on speech due to the importance of dereverberating speech for downstream analysis. VIDA consists of two main components (Figure 3): 1) a Visual Acoustics Network (VAN), which learns to map RGB-D images of the environment to features useful for dereverberation, and 2) the dereverberation module itself, which is based on a UNet encoder-decoder architecture. The UNet encoder takes as input a reverberant spectrogram, while the decoder takes the encoder's output along with the visual dereverberation features produced by the VAN and reconstructs a dereverberated version of the audio.

**Visual Acoustics Network.** Visual observations of a scene reveal information about room acoustics, including room geometry, materials, object locations, and the speaker position. We devise the VAN to capture all these cues into a latent embedding vector, which is subsequently used to remove reverb. This network takes as its input an RGB image $I_r$ and a depth image $I_d$, captured from the listener's current position within the environment. The depth image contains information about the geometry of the environment and arrangement of objects, while the RGB image contains more information about

their material composition. To better model these different information sources, we use two separate ResNet18 (He et al., 2016) networks to extract their features, i.e. $e_r = f_r(I_r)$ and $e_d = f_d(I_d)$. We concatenate $e_r$ and $e_d$ channel-wise and feed the result to a 1x1 convolution layer $f_c(\cdot)$ to reduce the number of total channels to 512 followed by a subsequent pooling layer $f_l(\cdot)$ to reduce the spatial dimension, resulting in the output vector $e_c = f_l(f_c([e_r; e_d]))$.

**Dereverberation Network.** To recover the clean speech audio, we use the UNet (Ronneberger et al., 2015) architecture, a fully convolutional network often used for image segmentation. We first use the Short-Time Fourier Transform (STFT) to convert the reverberant input audio $A_r$ to a complex spectrogram $S_r$. We treat $S_r$ as a 2-dimensional, 2-channel image, where the horizontal dimension represents time, the vertical dimension represents frequency, and the two channels represent the log-magnitude and phase angle. Our UNet takes spectrograms of a fixed size of $256 \times 256$ as input, but in general the duration of the speech audio we wish to dereverberate will be variable. Therefore, the model processes the full input spectrogram using a series of overlapping, sliding windows. Specifically, we segment the spectrogram along the time dimension into a sequence of fixed-size chunks $S_r^{seg} = \{S_r^1, S_r^2, ..., S_r^n\}$ using a sliding window of length $s$ frames and 50% overlap between consecutive windows to avoid boundary artifacts. To derive the ground-truth target spectrograms used in training, we perform the exact same segmentation operation on the clean source audio $A_s$ to obtain $S_s^{seg} = \{S_s^1, S_s^2, ..., S_s^n\}$.

During training, when a particular waveform $S_r$ is selected for inclusion in a data batch, we randomly sample one of its segments $S_r^i$ to be the input to the model, and choose the corresponding $S_s^i$ as the target. We first compute the output of the VAN, $e_c$, for the environment image associated with $S_r$. Next, $S_r^i$ is fed to the UNet's encoder to extract the intermediate feature map $e^s = f_{enc}(S_r^i)$. We then spatially tile and concatenate $e^c$ depth-wise with $e^s$, and feed the fused features to the UNet decoder, which predicts the source spectrogram segment $\hat{S}_s^i = f_{dec}([e^s, e^c])$.

**Spectrogram prediction loss.** The primary loss function we use to train our model is the Mean-Squared Error (MSE) between the predicted and ground-truth spectrograms, treating the magnitude and phase separately. For a given predicted spectrogram segment $\hat{S}_s^i$, let $\hat{M}_s^i$ denote the predicted log-magnitude spectrogram, $\hat{P}_s^i$ denote the predicted phase spectrogram, and $M_s^i$ and $P_s^i$ denote the respective ground-truth magnitude and phase spectrograms. We define the magnitude loss as:

$$L_{magnitude} = ||M_s^i - \hat{M}_s^i||_2. \tag{1}$$

To address the issue of phase wraparound, we map the phase angle to its corresponding rectangular coordinates on the unit circle and then compute the MSE loss for the phase:

$$L_{phase} = ||\sin(P_s^i) - \sin(\hat{P}_s^i)||_2 + ||\cos(P_s^i) - \cos(\hat{P}_s^i)||_2. \tag{2}$$

**Reverb-visual matching loss.** To reinforce the consistency between the visually-inferred room acoustics and the reverberation characteristics learned by the UNet encoder, we also employ a contrastive reverb-visual matching loss:

$$L_{matching}(e^c, e^s, e_n^s) = \max\{d(f_n(e^c), f_n(e^s)) - d(f_n(e^c), f_n(e_n^s)) + m, 0\}. \tag{3}$$

Here, $d(x, y)$ represents L2 distance, $f_n(\cdot)$ applies L2 normalization, $m$ is a margin, and $e_n^s$ is a different speech embedding sampled from the same data batch. This loss forces the embeddings output by the VAN and the UNet encoder to be consistent, which we empirically show to be beneficial.

**Training.** Our overall training objective (for a single training example) is as follows:

$$L_{total} = L_{magnitude} + \lambda_1 L_{phase} + \lambda_2 L_{matching}, \tag{4}$$

where $\lambda_1$ and $\lambda_2$ are weighting factors for the phase and matching losses. To augment the data, we further choose to rotate the image view for a random angle for each input during training. This is possible because our audio recording is omni-directional and is independent of camera pose. This data augmentation strategy prevents the model from overfitting; without it our model fails to converge. This strategy creates a one-to-many mapping between reverb and views, forcing the model to learn a viewpoint-invariant representation of the room acoustics.

**Testing.** At test time, we wish to re-synthesize the entire clean waveform instead of a single fixed-length segment. In this case, we feed all of the segments for a waveform $S_r$ into the model and temporally concatenate all of the output segments. Because consecutive segments overlap by 50%, during the concatenation step we only retain the middle 50% of the frames from each segment and discard the rest. Finally, to re-synthesize the waveform we use the Griffin-Lim algorithm (Griffin &

| | Speech Enhancement | Speech Recognition | | Speaker Verification | |
|---|---|---|---|---|---|
| | PESQ ↑ | WER (%) ↓ | WER-FT (%) ↓ | EER (%) ↓ | EER-FT (%) ↓ |
| Anechoic (Upper bound) | 4.64 | 2.50 | 2.50 | 1.62 | 1.62 |
| Reverberant | 1.54 | 8.86 | 4.62 | 4.69 | 4.57 |
| MetricGAN+ (Fu et al., 2021) | 2.33 (+51%) | 7.49 (+15%) | 4.86 (-5%) | 4.67 (+0.4%) | 2.75 (+39%) |
| HiFi-GAN (Su et al., 2020b) | 1.83 (+19%) | 9.31 (-5%) | 5.59 (-20%) | 4.30 (+8%) | 2.49 (+46%) |
| WPE (Nakatani et al., 2010) | 1.63 (+6%) | 8.18 (+8%) | 4.30 (+7%) | 5.19 (-11%) | 4.48 (+2%) |
| Audio-only dereverb. | 2.32 (+51%) | 4.92 (+44%) | 3.76 (+19%) | 4.67 (+0.4%) | 2.61 (+43%) |
| VIDA w/ normal FoV | 2.33 (+51%) | 4.85 (+45%) | 3.73 (+19%) | 4.53 (+3%) | 2.79 (+39%) |
| VIDA w/o matching loss | **2.38** (+55%) | 4.59 (+48%) | 3.72 (+19%) | 4.02 (+14%) | 2.62 (+43%) |
| VIDA w/o human mesh | 2.31 (+50%) | 4.57 (+48%) | 3.72 (+19%) | 4.00 (+15%) | 2.52 (+45%) |
| VIDA w/ random image | 2.34 (+52%) | 4.94 (+44%) | 3.82 (+17%) | 4.70 (-0.2%) | 2.48 (+2%) |
| VIDA | 2.37 (+54%) | **4.44** (+50%) | **3.66** (+21%) | **3.97** (+15%) | **2.40** (+47%) |

Table 1: Results on multiple speech analysis tasks, evaluated on the LibriSpeech test-clean set that is reverberated with our environmental simulator (with the exception of the "Anechoic (Upper bound)" setting, which is evaluated on the original audio). WER-FT and EER-FT refer to tests where the ASR and verification models are finetuned with the audio-enhanced data, respectively. The relative improvement compared to Reverberant is included in parentheses.

Lim, 1984) to iteratively improve the predicted phase for 30 iterations, which we find works better than directly using the predicted phase or using Griffin-Lim with a randomly initialized phase.

## 6 EXPERIMENTS

We evaluate our model by dereverberating speech for three downstream tasks: speech enhancement (SE), automatic speech recognition (ASR), and speaker verification (SV). We evaluate using both real scanned Matterport3D environments with sim audio as well as real-world data collected with a camera and mic. Please see Supp. for all hyperparameter settings and data preprocessing details.

**Evaluation tasks and metrics.** We report the standard metrics Perceptual Evaluation of Speech Quality (PESQ) (Rix et al., 2001), Word Error Rate (WER), and Equal Error Rate (EER) for the three tasks, respectively. For ASR and SV, we use pretrained models from the SpeechBrain (Ravanelli et al., 2021) toolkit. We evaluate these models off-the-shelf on our (de)reverberated version of the LibriSpeech test-clean set, and also explore finetuning the model on the (de)reverberated LibriSpeech train-clean-360 data. For performing speaker verification, we construct a set of 80k sampled utterance pairs consisting of different rooms, mic placements and genders to account for session variability, similar to Richey et al. (2018). Please see Supp. for more details.

**Baseline models.** In addition to evaluating the anechoic and reverberant audio (with no enhancement), we compare against multiple baseline dereverberation models: 1. **MetricGAN+** (Fu et al., 2021): a state-of-the-art model for speech enhancement; we use the public implementation from SpeechBrain (Ravanelli et al., 2021), trained on our dataset. Following the original paper, we optimize for PESQ during training, then choose the best-performing model snapshot (on the validation data) specific to each of the downstream tasks. 2. **HiFi-GAN** (Su et al., 2020b): a recent model for denoising and dereverberation.[1] 3. **WPE** (Nakatani et al., 2010): A widely used statistical speech dereverberation model. 4. **Audio-only dereverb**: An ablation of VIDA that does not use any visual input or the proposed matching loss (i.e., the VAN is removed). It uses only the U-Net trained with the MSE loss for dereverberation; a similar model is proposed by Ernst et al. (2018).

We emphasize that all baselines are *audio-only* models, as opposed to our proposed *audio-visual* model. Our multimodal dereverberation technique could extend to work in conjunction with other newly-proposed audio-only models, i.e., ongoing architecture advances are orthogonal to our idea.

**Results in scanned environments.** Table 1 shows the results for all models on SE, ASR, and SV. First, since existing methods report results on anechoic audio, we note the pretrained SpeechBrain model applied to anechoic audio (first row) yields errors competitive with the SoTA (Gulati et al., 2020), meaning we have a solid experimental testbed. Comparing the results on anechoic vs. reverberated speech, we see that reverberation significantly degrades performance on all tasks. Our VIDA model outperforms all other models, and by a large margin on the ASR and SV tasks: without

---

[1]We use this implementation: `https://github.com/rishikksh20/hifigan-denoiser`.

|  | Speech Enhancement PESQ ↑ | Speech Recognition WER (%) ↓ | Speaker Verification EER (%) ↓ |
|---|---|---|---|
| Anechoic (Upper bound) | 4.64 | 2.52 | 1.42 |
| Reverberant | 1.22 | 18.39 | 3.91 |
| MetricGAN+ (Fu et al., 2021) | **1.62** (+33%) | 21.42 (-16%) | 5.70 (-46%) |
| HiFi-GAN (Su et al., 2020b) | 1.33 (-9%) | 24.05 (-31%) | 5.21 (-33%) |
| Audio-only dereverb. | 1.41 (+16%) | 15.18 (+17%) | 4.24 (-8%) |
| VIDA w/ normal FoV | 1.44 (+18%) | 14.71 (+20%) | 3.79 (+3%) |
| VIDA | 1.49 (+22%) | **13.02** (+29%) | **3.75** (+4%) |

Table 2: Results on real data demonstrating sim2real transfer. Gain vs. Reverberant in parens.

|  | Atrium | Auditorium | Meeting Room | Classroom | Corridor |
|---|---|---|---|---|---|
| Near-field (1-3m) | 14.10 / **8.97** | **0.91** / **0.91** | **4.98** / 6.47 | 6.14 / **5.26** | 2.15 / **1.79** |
| Mid-field (3-8m) | 21.78 / **18.94** | **5.06** / 6.32 | **7.67** / **7.67** | 2.56 / **1.47** | 7.27 / **4.36** |
| Far-field (8+m) | 52.38 / **50.52** | 10.44 / **7.46** | 21.95 / **6.71** | **5.91** / 6.82 | 25.23 / **21.10** |

Table 3: Breakdown of word error rate (WER) for Audio-only dereverb. / VIDA on real test data.

finetuning, we achieve absolute improvements of 0.04 PESQ (1.71% relative improvement), 0.48% WER (9.75% relative improvement), and 0.68% EER (14.56% relative improvement) over the *best baseline* in each case (which happens to be the audio-only version of VIDA for both the ASR and SV tasks). The results are statistically significant according to a paired t-test (p-values are 1.56e-60 for PESQ, 3.70e-08 for WER, 2.58e-43 for speaker verification scores). After finetuning the ASR and SV models, the gains are still largely preserved at 0.1% WER (2.66% relative), and 0.21% EER (8.03% relative), although it is important to note that finetuning downstream models on enhanced speech is not always feasible, e.g., if using an off-the-shelf ASR system. Our results demonstrate that learning the acoustic properties of an environment from visual signals is very helpful for dereverberating speech, enabling the model to leverage information unavailable in the audio alone.

To understand how the WER of our model changes as the input difficulty increases, we plot the cumulative WER against (a) the PESQ of the input speech and (b) the speaker distance from the camera in Fig. 4a and Fig. 4b, respectively. Our VIDA model consistently outperforms the Audio-only baseline and MetricGAN+ (Fu et al., 2021) on all difficulty levels. Importantly, when the input sample is more difficult (low PESQ or far distance), our model shows a greater performance gain.

**Ablations.** To understand how well VIDA works with a normal field-of-view (FoV) camera, we replace the panorama image input with a FoV of 80 degrees randomly sampled from the current view. Table 1 shows the results. All metrics drop compared to using a panorama. This is expected, because the model is limited in what it can see with a narrower field of view; the inferred room acoustics are impaired by not seeing the full environment or missing where the speaker is. Compared to the audio-only dereverberation model, however, VIDA still performs better; even a partial view of the environment helps the model understand the scene and dereverberate the audio.

Next, we ablate the proposed reverb-visual matching loss ("w/o matching loss"). Without it, VIDA's performance declines on all metrics. This shows by forcing the visual feature to agree with the reverberation feature, our model learns a better representation of room acoustics.

To examine how much the model leverages the human speaker cues and uses the visual scene, we evaluate our trained VIDA model on the same test data but with the 3D humanoid removed ("w/o human mesh") or train VIDA with random images ("w/ random image") and evaluate. All three metrics become worse. This shows our model pays attention to both the presence of the human speaker and the scene geometry to learn a better representation of the anticipated reverberation.

Lastly, to understand if accurately estimating room acoustics characteristics (e.g. RT60) is enough for dereverberation, we equip the audio-only dereverb baseline with predicted room acoustics characteristics and show that these models still underperform VIDA. See details in Supp.

**Robustness to noise and ambient sounds.** We test the robustness of our model by adding ambient sounds from environments such as coffee shops and restaurants, using the WHAM dataset (Wichern et al., 2019). We add them to the reverberant test waveform with a 20dB SNR, following (Ernst et al., 2018; Nakatani et al., 2010). We show our model is robust to this noise; see Supp.

**Results on real data.** Next, we deploy our model in the real world. We use all models trained in simulation to dereverberate the real-world dataset (cf. Sec. 4) before using the finetuned ASR/SV

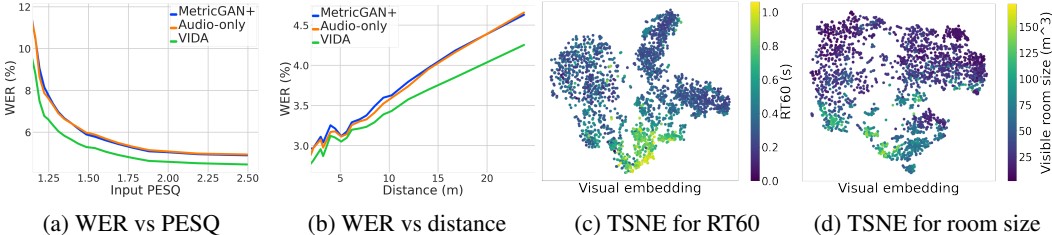

|  (a) WER vs PESQ  |  (b) WER vs distance  |  (c) TSNE for RT60  |  (d) TSNE for room size  |

Figure 4: Left: Cumulative WER against PESQ of the input speech (a) and the speaker distance. from the camera (b). Right: TSNE visual features colored by RT60 (c) and visual room size (d).

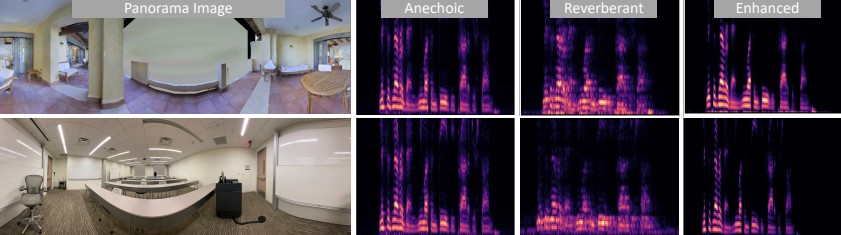

Figure 5: Example input images, anechoic spectrograms, reverberant spectograms and spectrograms dereverberated by VIDA (top is from a scan, bottom is a real pano). The speaker is out of view in the first case and distant in the second case (back of the classroom). Though both received audio inputs are quite reverberant, our model successfully removes the reverb and restores the source speech.

models to evaluate the enhanced speech. Table 2 shows the results of all models on real data. Reverberation does more damage to the WER compared to in simulation. Although MetricGAN+ (Fu et al., 2021) has surprisingly better PESQ, it has a weak WER score. Our VIDA model again outperforms all baselines on ASR and SV. This demonstrates the realism of the simulation and the capability of our model to transfer to real-world data, a promising step for VIDA's wider applicability.

Table 3 breaks down the ASR performance for Audio-only dereverb. (the best baseline) and our VIDA model by environment type and speaker distance. The atrium is quite reverberant due to the large space. Although the auditorium is similarly large, the space is designed to reduce reverberation and thus both models have lower WER. The meeting room and the classroom have smaller sizes and are comparatively less reverberant. The corridor only becomes reverberant when the speaker is far away. VIDA outperforms the Audio-only dereverb. baseline in most cases, especially in highly reverberant ones. See the Supp. video for examples.

**Qualitative examples.** Figure 5 shows a simulated and real-world example. As we can see, the reverberant spectrogram is much blurrier compared to the anechoic spectrogram, while our predicted spectrogram removes those reverberations by leveraging the visual cues of room acoustics.

**Analyzing learned visual features.** Figure 4c and 4d analyze our model's learned visual features via 2D TSNE projections (van der Maaten & Hinton, 2008). For each sample, we color the point according to either (c) RT60 or (d) visible room volume from the camera's viewpoint calculated based on the depth map (depth values truncated at 10m). Neither of these variables are available to our model during training, yet when learning to perform deverberation, our model exposes these high-level properties relevant to the audio-visual task. Consistent with the quantitative results above, this analysis shows how our model captures elements of the visual scene, room geometry, and speaker location that are valuable to proper dereverberation.

## 7 CONCLUSION

We introduced the novel task of audio-visual dereverberation. The proposed VIDA approach learns to remove reverb by attending to both the audio and visual streams, recovering valuable signals about room geometry, materials, and speaker locations from visual encodings of the environment. In support of this task, we develop a large-scale dataset providing realistic, spatially registered observations of speech and 3D environments. VIDA successfully dereverberates novel voices in novel environments more accurately than an array of baselines, improving multiple downstream tasks. Trained in a simulator, our model also shows promise enhancing speech in real-world data, though more work is needed to bring down the absolute error rates. In future work, we will explore temporal models for dereverberation with real-world video and introduce active perception for camera control.

REPRODUCIBILITY

We provide the training details in Sec. 5, implementation details (hyperparameters and architectures) in Sec. A.3 and evaluation details in Sec. A.4. For dataset creation, details can be found in Sec. 2. Both the code and dataset will be made public if accepted.

ETHICS STATEMENT

The research in this paper does not involve human subjects. The people in the real data have consented to share their imagery. Our datasets—both the simulations and those images—can be shared publicly. All authors of this paper follow the ICLR Code of Ethics (https://iclr.cc/public/CodeOfEthics).

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

# A    SUPPLEMENTARY MATERIALS

In this supplementary material, we provide additional details about:

1. Video (with audio) for demos of the collected data as well as qualitative assessment of VIDA's performance.
2. Distribution of average RT60 over room types.
3. Implementation details of our model and data pre-processing.
4. Evaluation details of downstream tasks and corresponding metrics.
5. Robustness to noise and ambient sound.
6. Predicting room acoustics characteristics.
7. Ablation on visual sensors.
8. Cumulative WER against PESQ and distance.

## A.1    QUALITATIVE VIDEO

This video includes examples for audio-visual data in simulation and in the real-world. We demonstrate examples of our dereverbration model applied to these inputs.

## A.2    RT60 DISTRIBUTION

Table 4 shows the distribution of average RT60 over some common room types in the Matterport3D training environments. Different rooms have different room acoustics characteristics, associated with their room sizes, typical object arrangements and materials. Importantly, those properties are visually detectable. For example, bedrooms are usually smaller than garage or lounge, and have beds with absorptive materials. The joint effects of these factors lead to small reverberation time. This knowledge is what models could learn to leverage for understanding room acoustics and performing dereverberation better.

|      | Bedroom | Dining Room | Hallway | Living Room | Garage | Lounge | Conference Room |
|------|---------|-------------|---------|-------------|--------|--------|-----------------|
| RT60 | 0.211s  | 0.224s      | 0.239s  | 0.242s      | 0.255s | 0.318s | 0.393s          |

Table 4: Distribution of average RT60 when the speaker is 1m away from mic.

## A.3    IMPLEMENTATION DETAILS

For the STFT calculation, we sample the input audio at 16 kHz and use a Hamming window of size 400 samples (25 milliseconds), a hop length of 160 samples (10 milliseconds), and a 512-point FFT. By retaining only the positive frequencies and segmenting the spectrograms into 256-frame chunks (corresponding to approximately 2.5 seconds of sound), the final audio input size to our U-Net is 256x256. We use the Adam optimizer (Kingma & Ba, 2015) to train our model with $lr = 0.001$. We decay the learning rate exponentially to $lr = 0.0001$ in 150 epochs. We set the batch size to 96 and train all models for 150 epochs, which is long enough to reach convergence. We set the margin $m$ to 0.5, phase loss weight $\lambda_1$ to 0.08 and matching loss weight $\lambda_2$ to 0.001.

## A.4    EVALUATION DETAILS

We evaluate our model on three tasks: speech enhancement (SE), automatic speech recognition (ASR), and speaker verification (SV).

- For SE, the goal is to improve the overall sonic quality of the speech signal, which we measure automatically using the standard Perceptual Evaluation of Speech Quality (PESQ) (Rix et al., 2001) metric.

- For ASR, the goal is to automatically transcribe the sequence of words that were spoken in the audio recording. For this task, we report the Word Error Rate (WER), which is the standard metric used in ASR and reflects a word-level edit distance between a recognizer's output and the ground-truth transcription.

- For SV, the goal is to detect whether or not two different spoken utterances were spoken by the same speaker. For SV, we report the Equal Error Rate (EER), a standard metric in the SV field indicating the point on the Detection Error Tradeoff (DET) curve where the false alarm and missed detection probabilities are equivalent.

|  | Speech Enhancement | Speech Recognition | | Speaker Verification | |
|---|---|---|---|---|---|
|  | PESQ ↑ | WER (%) ↓ | WER-FT (%) ↓ | EER (%) ↓ | EER-FT (%) ↓ |
| Anechoic (Upper bound) | 4.64 | 2.50 | 2.50 | 1.62 | 1.62 |
| Reverberant | 1.36 | 12.27 | 6.38 | 4.69 | 5.10 |
| MetricGAN+ (Fu et al., 2021) | **2.12** (+57%) | 9.40 (+23%) | 7.09 (-11%) | 4.94 (-5%) | 3.38 (+34%) |
| HiFi-GAN (Su et al., 2020b) | 1.55 (+14%) | 11.90 (+3%) | 7.80 (%) | 5.18 (-10%) | 3.80 (+25%) |
| WPE (Nakatani et al., 2010) | 1.39 (+2%) | 11.32 (+8%) | 7.00 (-10%) | **4.48** (+4%) | 4.95 (+3%) |
| Audio-only dereverb. | 1.76 (+29%) | 7.37 (+40%) | 5.52 (+14%) | 5.75 (-23%) | 3.58 (+30%) |
| VIDA w/ normal FoV | 1.76 (+29%) | 7.51 (+39%) | 5.51 (+14%) | 5.54 (-18%) | 3.40 (+33%) |
| VIDA w/o matching loss | 1.81 (+33%) | 6.76 (+45%) | 5.31 (+17%) | 4.95 (-6%) | 3.26 (+36%) |
| VIDA | 1.82 (+34%) | **6.53** (+47%) | **5.29** (+17%) | 4.83 (-3%) | **3.13** (+39%) |

Table 5: Results on the LibriSpeech test-clean set mixed with ambient sounds at a 20 dB signal-to-noise ratio.

Since the spectrogram MSE loss we optimize during training does not perfectly correlate with these three task-specific metrics, we perform model selection (across snapshots saved each training epoch) by computing the task-specific evaluation metric on 500 validation samples. We then select the best model snapshot independently for each downstream task and evaluate on the held-out test set; the same model selection procedure is also used for all of our baseline models.

For the ASR and SV tasks, we use the SpeechBrain (Ravanelli et al., 2021) toolkit. For ASR, we use the HuggingFace Transformer (Vaswani et al., 2017) + Transformer LM model pre-trained on LibriSpeech (Panayotov et al., 2015). We evaluate this model off-the-shelf on our (de)reverberated version of the LibriSpeech test-clean set, and also explore fine-tuning the model on the (de)reverberated LibriSpeech train-clean-360 data. For the SV task, we use SpeechBrain's ECAPA-TDNN embedding model (Desplanques et al., 2020), pre-trained on VoxCeleb (Arsha Nagrani, 2017). For performing verification, we evaluate the model on a set of 80k randomly sampled utterance pairs from the test-clean set (40k same-speaker pairs, 40k different-speaker pairs) using the cosine similarity-based scoring pipeline from SpeechBrain's VoxCeleb recipe. In the verification task, we use the clean (non-reverberated) speech as the reference utterance, and the reverberant speech as the test utterance. As in the ASR task, we evaluate this model on our dereverberation model's outputs both off-the-shelf, as well as after fine-tuning on the (de)reverberated train-clean-360 set.

## A.5  ROBUSTNESS TO NOISY AUDIO

We test the robustness of our model by adding ambient sounds from urban environments such as coffee shops, restaurants, and bars using the WHAM dataset (Wichern et al., 2019). We add them to the reverberant test waveform with a SNR of 20, following (Ernst et al., 2018; Nakatani et al., 2010). Table 5 shows the results on three downstream tasks. As expected, all models' performance drop compared to the results in the noise-free test setting (Table 1), but our VIDA model still significantly outperforms the baselines on ASR. For speaker verification, WPE (Nakatani et al., 2010) is reported to be robust to noisy input and thus has lower EER while using the pretrained model, but it underperforms VIDA when the SV model is finetuned on the enhanced speech. Noise has less impact on the performance on MetricGAN+(Fu et al., 2021) likely because it directly optimizes PESQ.

## A.6  PREDICTING ROOM ACOUSTICS CHARACTERISTICS

RT60 and direct-to-reverberant ratio (DRR) are some common characteristics used for measuring room acoustics (Giri et al., 2015). To understand if estimating these parameters accurately is enough for dereverberation, we first performed predictions of RT60, DRR, and distance purely based on audio with a ResNet network. After converging, the average absolute errors for them are 3e-3s, 160 dB and 17.92m respectively. It shows predicting DRR or distance is quite hard from audio alone. Next, we concatenate the predicted quantities as an extra channel of input to the audio latent feature and train these models with the same setup until convergence.

Table 6 shows the results. Even though these models leverage extra supervision for training that is not available to VIDA, they still underperform our VIDA model and only marginally improve the performance of the audio-only model. Among these three predicted quantities, we can see predicting distance helps the most, which is difficult to predict even supplied with ground truth, but is more predictable from the visual stream.

|  | Speech Enhancement PESQ ↑ | Speech Recognition WER (%) ↓ | Speaker Verification EER (%) ↓ |
|---|---|---|---|
| Reverberant | 1.54 | 8.86 | 4.69 |
| Vanilla Audio-only | 2.32 (+51%) | 4.92 (+44%) | 4.67 (+0.4%) |
| Audio-only w/ RT60 prediction | 2.36 (+53%) | 4.75 (+46%) | 4.67 (+7%) |
| Audio-only w/ DRR prediction | 2.34 (+52%) | 4.86 (+45%) | 4.51 (+4%) |
| Audio-only w/ distance prediction | 2.36 (+53%) | 4.74 (+47%) | 4.27 (+9%) |
| VIDA | **2.37** (**+54%**) | **4.44** (**+50%**) | **3.97** (**+15%**) |

Table 6: Augmenting Audio-only dereverb. with predicted room acoustics characteristics.

|  | Speech Enhancement PESQ ↑ | Speech Recognition WER (%) ↓ | Speaker Verification EER (%) ↓ |
|---|---|---|---|
| Reverberant | 1.54 | 8.86 | 4.69 |
| Audio-only dereverb. | 2.32 (+51%) | 4.92 (+44%) | 4.67 (+0.4%) |
| VIDA w/o RGB | **2.38** (**+55%**) | 4.76 (+46%) | **3.82** (**+19%**) |
| VIDA w/o depth | **2.38** (**+55%**) | 4.52 (+49%) | 3.99 (+15%) |
| VIDA w/ early fusion | **2.38** (**+55%**) | 4.56 (+48.5%) | 3.94 (+16%) |
| VIDA | 2.37 (+54%) | **4.44** (**+50%**) | 3.97 (+15%) |

Table 7: Ablations on visual sensors. Percentages in parenthesis represent relative improvements over the reverberant baseline.

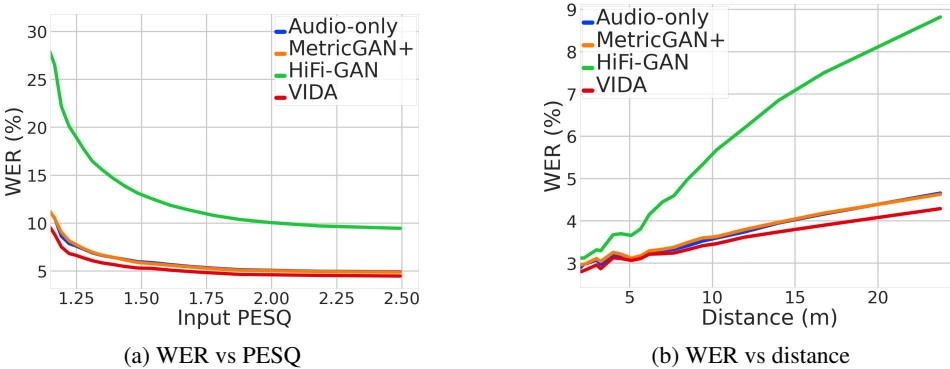

(a) WER vs PESQ    (b) WER vs distance

Figure 6: Cumulative WER against PESQ of the input speech (a) and the speaker distance.

## A.7 ABLATION ON VISUAL SENSORS

To understand the importance of each input sensor, we ablate the RGB and depth input as shown in Table 7. Dropping either RGB or depth makes the WER worse. We hypothesize that this is because they contain distinct information for the learning of room acoustics. The depth image is better for capturing room geometry, while the RGB image is better for capturing material and speaker location information.

In addition, we perform early fusion of RGB and Depth images by stacking them along the channel dimension (w/ early fusion in Table 7) and use one ResNet18 (He et al., 2016) model instead of two. This method also has worse WER, which validates our design choice of extracting RGB and depth features separately.

## A.8 CUMULATIVE WER AGAINST WER AND DISTANCE

Figure 6 shows the cumulative WER against PESQ of the input speech and the speaker distance from the camera in Fig. 6a and Fig. 6b respectively. In this plot, we include the curves for HiFi-GAN (Su et al., 2020b) as well, which underperforms audio-only dereverb., MetricGAN+ and VIDA by a large margin.

## A.9 PARAMETER COUNT AND EFFICIENCY.

In Table 8, we show the number of parameters and how long it takes to do one forward pass with batch size 1 on a RTX 6000 GPU for each model. The results are averaged over 2000 runs. Despite

|  | MetricGAN | HiFI-GAN | WPE | Audio-only | VIDA |
|---|---|---|---|---|---|
| Parameter Count | 1.9M | 42.6M | 0 | 17.7M | 43.7M |
| Inference time | 0.032s | 0.032s | 0.011s | 0.005s | 0.009s |

Table 8: Parameter count and inference time for each model.

|  | Speech Enhancement PESQ ↑ | Speech Recognition WER (%) ↓ | Speaker Verification EER (%) ↓ |
|---|---|---|---|
| Vanilla Audio-only | 2.32 | 4.92 | 4.67 |
| Audio-only + R-vectors | 2.23 | 5.23 | 4.82 |
| VIDA | **2.37** | **4.44** | **3.97** |

Table 9: Augmenting Audio-only dereverb. with trained R-vectors.

having more parameters, VIDA runs comparably to Audio-only (ablation of VIDA) and faster than MetricGAN+, HiFi-GAN and WPE.

## A.10 COMPARISON WITH R-VECTORS

R-vectors (Khokhlov et al., 2019) is a recently proposed approach that trains a room acoustics aware network in a self-supervised fashion and uses the feature of this network to help improve distant speech recognition. Since there is no public implementation of this work, we implement this approach based on descriptions in the paper. We train the network with 49,430 distinct RIRs, each convolved with 20 speech clips, and the test accuracy is 88%. When using this network for dereverberation, we tile and concatenate R-vectors with the audio features similar to the audio-visual concatenation and train the network similarly to audio-only dereverb. We refer to this model as audio-only dereverb. + R-vectors. Table 9 shows the results of this model in comparison to other models. This acoustics-aware vector does not improve the performance of our audio-only model.

