# OpenReview forum: "Learning Audio-Visual Dereverberation"
_ICLR.cc/2022/Conference — ICLR 2022 Submitted_

### Official Review · Reviewer_j5yj · 2021-10-25

**Correctness:** 4
**Technical Novelty And Significance:** 4
**Empirical Novelty And Significance:** 4
**Recommendation:** 8
**Confidence:** 4

**Main Review:**

This paper takes an interesting and novel approach to de-reverberation.
The paper is well written, and each component of the method is well justified.
The qualitative analysis (e.g. figure 4) is a nice touch, and helps shed some light on the behavior of the proposed method (and baselines).
That said, I do have a few questions and suggestions.

My primary criticism is in the reporting of the evaluations (tables 1, 2, 3).
Each setting is summarized by a single average score, and the "best" number is written in bold with no discussion given to the distribution of scores on the test data or the practical meaning of score differences.
This may give readers an inflated sense of performance here: is PESQ=2.38 really any different from PESQ=2.37? 2.33?  (Likewise for WER, EER, etc.)
There is a brief mention of astronomically low p-values (1.56e-60?) which I expect are primarily driven by the size of the test set, and not the size of the effect.
I would strongly encourage the authors to do a more responsible job of reporting their results: include some error bars, or some other indication of the spread of the data!

The proposed method takes an interesting approach to optimizing both magnitude and phase (eqs 1 and 2) for the output spectrogram.
The Griffin-Lim algorithm is then used to refine the phase estimate, with a vague statement of this working better than directly using predicted phase or GL with random initialization.
While I won't ask for more experiments on this, I think the point deserves a bit more explanation than what is given here, especially since there is no ablation study on the phase reconstruction loss.  (Nor is there any discussion of hyper-parameter tuning for equation 4 beyond the settings reported in the appendix.)


I found Figure 5 (qualitative examples) to be pretty difficult to parse, and I say this as a reader with (probably) above-average experience reading spectrograms.
With a bit of squinting I can see what the authors are getting at, but it would be easier to read if the spectrograms were A) larger, and B) plotted on a logarithmic frequency axis so that the low frequencies (where most of the action is) occupy more of the visual real estate.


Finally, it was nice to see the results in table 3, which show quite a strong effect on the performance of the system depending on differences in distance and room geometry.
Not much is said about the geometric differences in the environments though, either for real or simulated data.
In particular, it looks from figure 2 as though there may be configurations of speaker/listener that have no direct path, in which one might say that everything observed is "reverb".
If this is the case, it would be interesting to see how performance (table 1) differs (if at all) in the presence or absence of a direct path.



**Summary Of The Paper:**

This paper describes an audio de-reverberation method which integrates (panoramic, rgb+depth map) visual input with a spectrogram U-Net to estimate a de-reverberated spectrogram and recover the original clean signal.
The proposed method is compared to several baseline (audio-only) de-reverberation methods on speech enhancement, recognition, and speaker verification problems using both synthetic and real data.
Several ablations of the proposed method are compared, along with some quantitative error analysis investigating the effects of distance and environment.
The proposed method appears to consistently improve on speech recognition and speaker verification, and perform about on par with prior work on speech enhancement (as measured by PESQ).



**Summary Of The Review:**

Overall, I enjoyed this paper!  I think some of the reporting could be done better, but this is a minor complaint.

---

> ### Author Response · Authors · 2021-11-23
> **Response to reviewer j5yj**
>
> Thank you for your valuable feedback.
>
> **Q1: Lack of distribution of scores or practical meaning of score difference.**
> The input speech varies drastically in terms of reverberation or difficulty per test sample, and thus there is large variance for all models' scores. Instead, to show the spread of scores, we plot the cumulative WER vs PESQ or distance to show how the scores vary as a function of difficulty in Figure 4 (a-b). As for the score difference, we explain in the "Results in scanned environments" paragraph in Section 6 that the pretrained ASR and SV models we used yield errors competitive with the SoTA, and thus the relative improvement is considerable.
>
> **Q2: Why using phase prediction + GL refinement is better than GL + random initialization.**
> Empirically we find predicting phase followed by GL refinement works better than using random initialization. We hypothesize that using the predicted phase to initialize GL helps convergence as opposed to using a randomly initialized phase. The PESQ, WER degrades from 2.37, 4.44% to 2.27, 4.50% when using the random initialization. For equation 4, we tuned the weights factors for phase loss and matching loss to achieve the best performance.
>
> **Q3: Figure 5 is hard to parse.**
> Thank you for the suggestion. We have updated the figures.
>
> **Q4: Geometric differences in the environments.**
> The distribution of WER as a function of distance is shown in Figure 4 (b). When distances become larger, the speaker is more likely to be out of view, in which case VIDA also outperforms baselines to the largest extent. To sort the results from our submission in a way that answers the reviewer's question even more directly, the table below shows the performance of all methods when there is direct sound (left 2 columns) and when there is no direct sound (right 2 columns). There are 1621 direct cases, and 979 non-direct cases. When there is no direct signal, the inputs tend to be more reverberant and difficult. Regardless, our VIDA model consistently outperforms baselines.
>
> |             | PESQ         | WER          | PESQ | WER |
> | :---        |    :----:   |       ---: |   :----:   |          ---: |
> | Reverberant | 1.67| 4.38 | 1.32| 16.71|
> | MetricGAN+  | 2.60| 3.60 | 1.88| 6.85 |
> | HiFi-GAN    | 2.09| 4.82 | 1.40| 16.39|
> | Audio-only  | 2.58| 3.42 | 1.89| 7.32 |
> | VIDA        | **2.61**| **3.33** | **1.96**| **6.27** |

---

### Official Review · Reviewer_NeeT · 2021-11-02

**Correctness:** 3
**Technical Novelty And Significance:** 2
**Empirical Novelty And Significance:** 2
**Recommendation:** 6
**Confidence:** 5

**Main Review:**

Strengths:
- first multi-modal (audio-visual) approach applied to dereverberation
- paper is well organized and straightforward to follow
- solid experiments with ablation studies and in-depth analyses
- supplementary materials provide useful information not explained/described in the main text

Weaknesses:
- some of the key arguments are not sufficiently validated
- contributions are marginal and limited
- experiments and evaluation are not thorough

The key idea in this work is that visual input captured by a camera conveys information that characterizes room acoustics. With all the experiments and ablation studies, however, I don't believe that the authors provide sufficient information that support the aforementioned key hypothesis. The authors did take into account this matter in two ways: first, they included a reverb-visual matching loss term to penalize a matching between random speech and visual input, but little experiment or analysis was performed on this loss term - varying the weight or using different loss functions, for example. With or without using a reverb-visual matching loss was included in the ablation studies, but the effect was not significant compared with other methods. The performance was better with the audio-only model even in speech quality metric (PESQ). The second way is to use a random image instead of using the image that matches the room impulse response. In such scenario, the images does not carry any information relevant to the room acoustics, and thus the performance should be worse than or comparable to that of the audio-only method. However, the performance is better in general and is even best for some metric. Further experiments and deeper analysis are warranted to investigate this issue.

Another point that I hope to be added is the comparison of the model efficiency. Adding a visual embedding network increases the number of model parameters and computational costs. The proposed method is likely to work in real-time scenarios, and possibly on small devices, the efficiency should also be considered.

Lastly, I'd like to suggest to include R-vectors approaches as another baseline for comparison.

**Summary Of The Paper:**

This paper presents a deep neural net-based dereverberation algorithm that uses both audio and video modalities. Based on the observation that a visual scene captured by a camera conveys information that is related to room characteristics, the authors propose a visually informed audio dereverberation method that aims to extract clean, anechoic speech from reverberant speech. In doing so, they first construct a large audio-visual dataset synthesized using a 3D simulator for real-world scanned environments and LibriSpeech data. They then train deep neural networks that take as inputs both visual data (RGB and depth images) and audio data (reverberant speech), and output clean speech. When training, two types of losses - one for clean speech spectrogram estimation and the other for reverb-visual matching - are used. Through the experiments with several downstream tasks for speech, they showed that the proposed audio-visual dererberation method outperforms the baseline models, both for synthetic and real-world test data.

**Summary Of The Review:**

This paper proposes a multi-modal learning framework that is applied to speech dereverberation. The idea is interesting and somewhat novel, but the experimental validation is not thorough enough to support the key idea of reverb-visual matching nor to provide sufficient evidences that visual stream does convey information about room acoustics.

---

> ### Author Response · Authors · 2021-11-23
> **Response to reviewer NeeT**
>
> Thank you for your valuable feedback.
>
> **Q1: Not sufficient information to support that visual inputs characterize room acoustics.**
> This is the first work that proposes to leverage visual inputs to characterize room acoustics and help dereverberation. We show that our audio-visual approach improves dereverberation on three speech tasks compared to multiple audio-only approaches, as experimentally validated on both synthetic and real data (Table 1, Table 2). These numbers strongly suggest that the model learns acoustics information from visuals to help dereverberation. Further, we analyze how the visual factors contribute to VIDA's performance: 1) full observation of the room geometry is useful to capture more geometric information (Table 1); 2) the model leverages the distance to speaker for understanding room acoustics ("w/o human mesh" ablation in Table 1); 3) the reverb-visual matching loss helps the model to learn better visual features; 4) we analyze the learned visual features by performing TSNE projection and show that they have good correlation with RT60 and room size (Figure 4 (c-d)).
>
> **Q2: Reverb matching loss, ``performance was better with the audio-only model even in PESQ".**
> The reviewer might have seen the wrong numbers. In Table 1, VIDA without reverb matching loss underperforms the full model, but still outperforms Audio-only on all metrics (note: higher PESQ is better). Secondly, while reverb-visual matching is one interesting part of our approach design, it is not the key idea of this paper. Our key idea is to leverage visual for dereverberation and we investigated ways to make model learn acoustics information better. Besides the matching loss, there are many other important factors that contribute to VIDA's performance (see the response to Q1).
>
> **Q3: Random image has pretty good performance compared to audio-only.**
> Thank you for raising this point.  As noted in the paper, this "w/ random image" setting was only used for testing the model (see para "Ablations" in Section 6).  It is likely that because our VIDA model is better trained, it has better performance than audio-only even when images are random at test time. Simply swapping the image with an incorrect image leads to a large drop on all three metrics, which indicates that our model relies on the acoustics cues in the image for dereverberation. However, taking the reviewer's feedback into account, we have now **trained** the model with random images and tested it.  The results are shown in the following table. When VIDA is trained with random images as input, its performance is very similar to the audio-only baseline, as the reviewer was expecting to see. This new ablation has exactly the same number of parameters and architecture as VIDA, and the only difference is the content of the image.  It shows that our model reasons about the image content to capture room acoustics to help dereverberation.
>
> |                           | PESQ                  | WER (%)   | EER (%)
> | :---                      |   :----:              |          ---: |          ---: |
> | Audio-only               | 2.32       | 4.92     | 4.67
> | Audio-only + R-vectors   | 2.23       | 5.23     | 4.82
> | VIDA w/ random image      | 2.34       | 4.94     | 4.70
> | VIDA                      | **2.37**       | **4.44**     | **3.97**
>
> **Q4: Model efficiency.**
> We updated the trainable parameter counts of all models and their inference time in Section A.9 of the appendix due to space constraints.  Despite having more parameters, VIDA runs comparably to Audio-only (ablation of VIDA) and faster than MetricGAN+, HiFi-GAN and WPE. We have updated Section 6 accordingly.
>
> **Q5: R-vectors approach for comparison.**
> Thank you for suggesting this baseline (Khokhlov et al., INTERSPEECH 2019). R-vectors proposes to train an acoustics-aware network in a self-supervised way and use its feature embedding for distant speech recognition. Though we could not find an existing public implementation of the R-vectors model suggested by the reviewer, we implemented and trained it on our dataset, where the R-vectors features are concatenated with the audio features similar to visual features. The result for this model is shown in the above table.  Adding R-vectors did not improve the performance of the Audio-only baseline. Due to the constraint of space, we have added this experiment and implementation details of R-vectors in Section A.10.

---

> > ### Comment · Reviewer_NeeT · 2021-12-05
> > **Changing my score from 5 to 6**
> >
> > I'd like to thank the authors for taking time to perform extra experiments to provide more information. Some of my concerns are cleared and thus I'm updating the score from 5 to 6 (marginally above the acceptance threshold). I still don't believe the novelty and performance improvement are significant.

---

> > > ### Author Response · Authors · 2021-12-06
> > > **Response to reviewer NeeT**
> > >
> > > Thank you very much for considering increasing your score! However, we haven’t seen the change reflected in the review. Could you kindly check if you have updated the initial review?

---

### Official Review · Reviewer_NGNE · 2021-11-02

**Correctness:** 4
**Technical Novelty And Significance:** 2
**Empirical Novelty And Significance:** 2
**Recommendation:** 3
**Confidence:** 4

**Main Review:**

While authors suggest visual information is helpful, it isn't reflected in the results. In Table 1, if we compare the "Audio-only dereverb." and "VIDA" results, the gains are very modest. The PESQ 2.32 to 2.37, WER-FT 3.76 to 3.66, and SID EER 2.61 to 2.40. This raises the question about the effectiveness of the proposed approach. I feel the majority of the dereverberation is still learned by the audio model. This should be investigated.

Overall, the technical contribution is limited to a new dataset and the addition of ResNet for visual information processing. Similar approaches have been used in the audio-visual enhancement with lip information.

**Summary Of The Paper:**

In this paper, the authors introduce a novel audio-visual dereverberation approach. They propose a Visually-Informed Dereverberation of Audio (VIDA) model for dereverberation. The authors also create synthetic/simulated datasets and real-world data for experimentation. Finally, they show the impact of the proposed VIDA model on several speech tasks including, recognition, enhancement, and speaker verification. The results are encouraging. The main contribution of this work is the use of visual information as an auxiliary input for dereverberation.





**Summary Of The Review:**

I believe the manuscript in its current form offers a very limited contribution to the research community, and the contribution of visual information in dereverberation is minimal.

---

> ### Author Response · Authors · 2021-11-23
> **Response to reviewer NGNE**
>
> Thank you for the valuable feedback.
>
> **Q1: Audio-only vs. VIDA gains are modest**
> We do not intend to claim that visuals can replace an audio model for dereverberation. Instead, we show that a visual model provides complementary information and leads to consistent performance gains. As for the performance difference, the relative improvement of VIDA over the audio-only baseline is 2% for PESQ, 10% for WER, and 15% for EER. The results are statistically significant according to a paired t-test (p-values are 1.56e-60 for PESQ, 3.70e-08 for WER, 2.58e-43 for speaker verification scores). We also validated the model's performance with a sim2real evaluation (Table 2, Table 3), which demonstrates VIDA is not simply learning to exploit artifacts of the simulator. For the significance of WER improvement, see discussion on LibriSpeech WER in W4 for reviewer Tgq4. Thus, our results do indeed reflect that visual information is helpful, contrary to the reviewer's claim.
>
> **Q2: Similar approaches have been used in audio-visual enhancement with lip information.**
> Unfortunately the reviewer writes no more than this single sentence about this point. The reviewer does not specify any references or elaborate in what way prior work conflicts with our claims about novelty. To our knowledge, no prior work attempts audio-visual dereverberation (see "Audio-visual learning from video" paragraph in Section 2). The visual information used by VIDA to infer reverberation is an observation of the room/environment.  Lip movements are not used in VIDA, though could be added as an additional, complementary visual input.

---

### Official Review · Reviewer_Tgq4 · 2021-11-03

**Correctness:** 4
**Technical Novelty And Significance:** 4
**Empirical Novelty And Significance:** 3
**Recommendation:** 6
**Confidence:** 5

**Main Review:**

### Strengths:

S1) To my knowledge, this is the first audio-visual approach to dereverberation. I think it is interesting to see how much visual information can help.

S2) The proposed dataset seems quite useful. A lot of existing synthetic reverb for such audio tasks is generated using shoebox room simulators, so having more realistic environments seems quite useful. The accompanying data recorded in real environments using loudspeaker playback (such that the clean reference is available) is also quite useful, and I hope the authors plan to release a recipe for this data.

S3) The paper includes a lot of ablations, which I really appreciate. Though the focus of the paper is on audio-visual approaches to the task, I'm really glad the paper evaluates audio-only versions of the model, as well as other components such as the matching loss.

S4) The paper is clear and well-written.

### Weaknesses:

W1) I listened to the demos in the supplementary material, and I feel like I heard a fair amount of artifacts and distortion in some examples: for moderate reverb, the model predictions have a certain "buzziness" to them, and more extreme levels of reverb have a "twanginess" to them. Something that would make this paper stronger would be a subjective human listening evaluation, e.g. MUSHRA, which would measure these kinds of artifacts better than an objective measure like PESQ. Also, alternative loss functions may help reduce artifacts. E.g. L1 loss instead of L2 on spectrogram magnitude, or MSE on power-compressed spectrograms can provide more perceptually-relevant loss functions.

W2) The processing method of using 2.56s frames, then concatenating the non-overlapping parts together seems suboptimal to me. Why not use e.g. a Hann window to stitch the predicted patches with overlap-add, or invert back to the time-domain and do overlap-add and train through this?

W3) Where does the "random speech embedding sampled from the data batch." come from? How is it computed? Is this a speaker ID embedding?

W4) The improvements by adding visual information are marginal compared to the audio-only model. The audio-only model does perform better than the baselines, but degrades PESQ compared to other baselines on real data. To benchmark the audio-only baseline against the literature, it would be interesting to compare to competitive systems in e.g. the REVERB challenge. I'm also not very convinced that visual input really helps that much.

W5) Table 1 would be improved by adding trainable parameter counts. Is the proposed model only doing better because it has more trainable parameters? WPE itself has no trainable parameters, since it's just a model-based EM algorithm.

W6) The lack of noise in the training and eval data limits real-world applicability, but I understand the focus of the paper in only on dereverberation. I'm glad that the authors tested noise robustness with WHAM! data, but 20 dB is a pretty high SNR for noise robustness. It would be interesting to see how performance degrades with decreasing SNR.

### Minor comments

M1) I feel that this statement isn't quite right: "R is a function of...the relative positioning of the speaker and the listener." I would argue that an RIR can be a function of the absolute positioning of the speaker and the listener. E.g. what if a source is close to a wall, versus in the the middle of a room?

M2) For the real data, would it be possible to be more specific about the source-to-microphone distances used in the real recordings, beyond "near-field to mid-field to far-field."?

M3) Also concerning real data: would it be possible to more more specific for this statement? "For each location, we play around 10 utterances.". How many utterances? Gender balanced? Different speakers across rooms?

M4) Nit: U-Net instead of UNet?

M5) "While our model is agnostic to the audio source type,": depends on the training data.

M6) In Tables 1 and 2, I would say "anechoic speech (upper bound)" instead of "clean speech (upper bound)", because to me "clean" suggests there's additive noise.

M7) About the blind RT60/DRR/distance predictions: "After converging, the average errors for them are 3e-3s, 160 dB and 17.92m respectively." Are these absolute or squared errors?

**Summary Of The Paper:**

This paper examines an audio-visual approach for dereverberation, where dereverberation of speech is conditioned on RGB and depth images (either field-of-view or panoramic). It proposes a dataset for this task based on real-world 3D scans of homes, using Librispeech data. Real data is also used The model is based on a U-Net conditioned on embeddings extracted by a "visual acoustics" network doing direct spectrogram prediction. The method is evaluated using PESQ, WER for speech recognition, and EER for speaker verification on synthetic and real data. The audio-visual method is found to perform marginally better than an audio-only version of the model. The audio-only version of the model outperforms several baselines from the literature on synthetic data, with mixed results on real data.

**Summary Of The Review:**

Overall, I vote for marginal acceptance. This is an interesting novel task (audio-visual dereverberation), and the proposed synthetic and real data are quite interesting and useful for the community. But the paper's results show that visual input provides marginal performance gain over the audio-only baseline. The audio-only baseline outperforms some benchmarks in terms of PESQ, WER for speech recognition, and EER for speaker verification on synthetic data, and WER and EER for real data, but I have concerns about the subjective listening quality of the demos, compared to other dereverberation algorithms I have listened to. Thus, I am concerned that this isn't a significant improvement over audio-only prior work, at least in terms of subjective listening quality.

---

> ### Author Response · Authors · 2021-11-23
> **Response to reviewer Tgq4**
>
> Thank you for the valuable feedback.
>
> **W1: Artifacts and distortion in some examples.**
> In this work, we focus on investigating if and how visuals help dereverberation for human perception (measured by PESQ) and machine perception (measured by WER and EER), and thus we perform spectrogram-level (frequency domain) optimization as it achieves a good balance between both. GAN-based models (MetricGAN+ and HiFi-GAN) produce audio with slightly less artifacts and distortion to our ears, but lead to significantly worse WER and EER scores (Table 1). Speech enhancement is only one of the 3 tasks we perform---and the only one of the 3 where human perception of the output is relevant (i.e., not so on speech recognition and speaker identification). We will explore a subjective user study for future work, but nonetheless the PESQ offers a measurable signal about the quality of our outputs relative to the baselines (W. Lin et al., Multimedia Analysis, Processing & Communications, 2011).
>
> **W2: Using a Hann window with overlap-add instead of non-overlapping concatenation.**
> Thank you for the suggestion. We tried overlap-add with Hann windows on spectrograms and time-domain signals, but neither of them improved the performance. We speculate that our non-overlapping concatenation works better because we directly minimize MSE loss on spectrograms.
>
>
> **W3: Where does the "random speech embedding sampled from the data batch." come from?**
> This random speech embedding is the audio feature of a different sample in the same input batch. We have updated Section 5 in the pdf.
>
> **W4: Marginal performance improvement by adding the visual information.**
> The relative improvement of VIDA over the audio-only baseline is 2% for PESQ, 10% for WER, and 15% for EER. The results are statistically significant according to a paired t-test (p-values are 1.56e-60 for PESQ, 3.70e-08 for WER, 2.58e-43 for speaker verification scores). The "w/ random images" ablation in Table 1 has exactly the same amount of parameters as VIDA---and it has performance similar to the audio-only model. The comparison between VIDA and this ablation strongly suggests that VIDA learns acoustic information to help dereverberation. We also validated the model's performance with a sim2real evaluation (Table 2, Table 3), which demonstrates VIDA is not simply learning to exploit artifacts of the simulator.
>
> We’d also like to point out that WER on LibriSpeech is in the low single digits, and absolute improvements of a fraction of a percent on any of the evaluation subsets are significant. The SOTA leaderboard for LibriSpeech is captured here: https://github.com/syhw/wer_are_we, only this doesn’t match our setting since these methods assume the input speech is clean. The “Voices from a Distance 2019 challenge” compares recent work on far-field, reverberant audio-only ASR: https://www.isca-speech.org/archive/Interspeech_2019/pdfs/1837.pdf. Among the top 4 systems (Figure 2), relative differences between adjacent systems range between 6.25% and 24.6%; we achieve a 11% relative gain over the audio-only baseline. The tasks are not identical (our method needs visual input and so is not applicable to Voices from a Distance) but this suggests that our gains are in a respectable zone for this community.
>
> **W5: Adding trainable parameter count to Table 1.**
> We added the trainable parameter counts for all models and their inference time in Section A.9 due to space constraints. We would argue that the performance improvement is not due to more trainable parameters because the "w/ random images" ablation in Table 1 has exactly the same amount of parameters as VIDA, which only performs similarly to audio-only.
>
> **W6: Adding noise with lower SNR.**
> In our submission, we had added noise with 20dB SNR following prior work (Ernst et al., 2018, Nakatani et al., 2010). Per the reviewer's request, to validate the model's robustness against extreme noise, we trained both audio-only and VIDA under WHAM noise with 5dB SNR. The PESQ, WER, EER scores for audio-only dropped from 2.33, 6.53, 4.83 to 1.52, 26.4 and 15.61. The PESQ, WER, EER scores for VIDA dropped from 2.37, 4.44, and 3.97 to 1.55, 25.37 and 14.16 respectively. Despite the huge drop of performance due to the extreme noise, our audio-visual model still outperforms the audio-only model on all tasks.

---

> > ### Author Response · Authors · 2021-11-23
> > **Response to reviewer Tgq4 (continued)**
> >
> > **M1: Minor: Inaccurate statement regarding RIRs.**
> > Thank you for pointing this out. We wanted to stress the relative positioning of the speaker with respect to the listener. But yes, RIR is a function of the absolute positioning of both the speaker and listener. We have updated Section 3.
> >
> > **M2: Be more specific about the source-to-microphone distances used in the real recordings.**
> > We have updated the distance range for each scenario in Table 3.
> >
> > **M3: Be more specific about the real data.**
> > We used 10 utterances balanced between the gender for each location. We use different speakers across rooms. We have incorporated this into the text in Section 4.
> >
> > **M7: Blind RT60/DRR/distance prediction accuracy.**
> > These errors are absolute. We have updated in Section A.6.
> >
> > **M4/M5/M6: Other minor writing comments.**
> > Thank you for your feedback. We have updated the pdf.

---

### Author Response · Authors · 2021-11-23
**Meta response for all reviewers**

We thank all reviewers for their valuable feedback. All reviewers acknowledge the novelty of developing the first audio-visual approach to dereverberation. Reviewers Tgq4, NGNE and NeeT appreciate the value of the created synthetic dataset and the collected real dataset. Reviewers Tgq4, NeeT and j5yj point out the solid experiments and ablations. They have also suggested some changes and asked for some clarifications. We address them in this rebuttal and by making minor revisions to the paper (highlighted in cyan).

---

### Decision · Program_Chairs · 2022-01-20

**Decision:**

Reject

**Comment:**

This paper investigates the dereverberation problem from the audio-visual perspective.  The geometry of the environment is represented by RGB and depth images.  The authors propose a so-called visually-informed dereverberation of audio (VIDA) model and also create a dataset consisting of both synthetic and real data to verify the effectiveness of the model.  Experiments are conducted on speech enhancement, speech recognition and speaker identification tasks.  The authors compare VIDA with audio only dereverberation as well as various established baseline systems in the community.

The audio-visual way of coping with dereverberation using visual representation of the acoustic environment seems to be interesting. The authors' rebuttal has cleared most of the concerns raised by the reviewers but there are still numerous lingering concerns which affect its acceptance.  First of all,  most of the reviewers consider the novelty not overwhelmingly significant.  Second, the contribution of the visual input seems to be only marginal compared to the audio-only dereverberation. Results on real data are also mixed.  Some of the reported p-values are extremely small, which raises questions whether it is due to the size of the test set.  Third, there are noticeable artifacts in some of the samples in the demo.  Fourth,  there are numerous issues in the paper that are worth further in-depth investigation. For instance, it would be helpful to show in which way exactly the RGB and depth images helps.